# Alcohol exposure disrupts mu opioid receptor-mediated long-term depression at insular cortex inputs to dorsolateral striatum

Braulio Muñoz[1], Brandon M. Fritz[1], Fuqin Yin[1] & Brady K. Atwood[1,2,3]

Drugs of abuse, including alcohol, ablate the expression of specific forms of long-term synaptic depression (LTD) at glutamatergic synapses in dorsal striatum (DS), a brain region involved in goal-directed and habitual behaviors. This loss of LTD is associated with altered DS-dependent behavior. Given the role of the μ-opioid receptor (MOR) in behavioral responding for alcohol, we explored the impact of alcohol on various forms of MOR-mediated synaptic depression that we find are differentially expressed at specific DS synapses. Corticostriatal MOR-mediated LTD (mOP-LTD) in the dorsolateral striatum occurs exclusively at inputs from anterior insular cortex and is selectively disrupted by in vivo alcohol exposure. Alcohol has no effect on corticostriatal mOP-LTD in dorsomedial striatum, thalamostriatal MOR-mediated short-term depression, or mOP-LTD of cholinergic interneuron-driven glutamate release. Disrupted mOP-LTD at anterior insular cortex–dorsolateral striatum synapses may therefore be a key mechanism of alcohol-induced neuroadaptations involved in the development of alcohol use disorders.

[1] Department of Psychiatry, Indiana University School of Medicine, Indianapolis, IN 46202, USA. [2] Department of Pharmacology and Toxicology, Indiana University School of Medicine, Indianapolis, IN 46202, USA. [3] Stark Neurosciences Research Institute, Indiana University School of Medicine, Indianapolis, IN 46202, USA. Correspondence and requests for materials should be addressed to B.K.A. (email: bkatwood@iu.edu)

Many drugs of abuse, including opioids and alcohol, impact neurotransmission in the dorsal striatum (DS)[1]. For example, we previously demonstrated that the heavily abused prescription opioid, oxycodone, ablates specific forms of synaptic plasticity in mouse DS[2]. While abuse of opioid analgesics is a growing and alarming epidemic[3], alcohol use disorders (AUDs) are also a significant source of great medical, social, and economic burdens throughout the world[4]. The progression of AUDs involves increasingly heightened activity within the basal ganglia in response to alcohol cues[5]. The DS is the primary input nucleus of the basal ganglia. Functionally, the DS can be divided into the dorsomedial striatum (DMS), which plays an important role in goal-directed learning, and the dorsolateral striatum (DLS), which regulates habit formation[6,7]. The specialized nature of these dorsal striatal subregions also extends to goal-directed and habitual ethanol (EtOH) seeking[7]. Both subregions are composed of ~95% GABAergic medium spiny projection neurons (MSNs) in rodents[8]. MSN activity is driven by glutamatergic input from three distinct sources: cerebral cortex, thalamic nuclei, and cholinergic interneurons (CINs) that can co-

**Fig. 1** MOR activation produces LTD of excitatory transmission in the dorsal striatum. **a** Schematic figure of coronal brain slice showing the recording of EPSCs evoked by focal electric stimulation in the dorsolateral striatum (DLS) of C57BL/6J mice. **b** Representative electrically evoked synaptic traces at baseline and after DAMGO (0.3 µM, 5 min) application. **c** DAMGO induced mOP-LTD of eEPSC amplitude in DLS MSNs of C57BL/6J mice ($n = 7$ slices from 4 mice). **d** The presence of mOP-LTD is not related to alterations in series resistance. **e** Schematic figure of coronal brain slice showing the recording of eEPSCs by focal electric stimulation in the dorsomedial striatum (DMS). **f** Representative electrically evoked synaptic traces before and after DAMGO (0.3 µM, 5 min) application. **g** The activation of MOR by DAMGO induced LTD of eEPSC amplitude in DMS MSNs of C57BL/6J ($n = 6$ from 2 mice). **h** The series resistance was stable during DMS recordings. Data represent mean ± SEM

release glutamate[9–11]. Drugs of abuse that affect glutamatergic input on to MSNs produce alterations in dorsal striatal-dependent behavior[12–17].

The opioid system is robustly expressed throughout the striatum[18] and is targeted by one of the few approved pharmacological interventions for AUDs, the opioid receptor antagonist naltrexone, which acts most selectively at μ-opioid receptors (MORs)[19]. MORs are key to modulating the rewarding effects of various drugs of abuse including alcohol[18]. The opioid system promotes synaptic plasticity in many brain regions[20–26], including the DS, where MOR activation induces static long-term depression (mOP-LTD) of glutamate release in both the DLS and DMS[2,27]. Interestingly, mOP-LTD is disrupted in the DLS by a single in vivo exposure to oxycodone, an effect persisting for 3 days post exposure[2]. In vivo EtOH exposure ablates endocannabinoid LTD (eCB-LTD), another form of LTD in the DS[13,15,17]. We previously showed that mOP-LTD and eCB-LTD interact and an oxycodone injection also interferes with eCB-LTD[2]. We hypothesized that EtOH would also disrupt mOP-LTD. However, it is unclear which glutamatergic input(s) express mOP-LTD. Our previous work demonstrated that inputs from motor cortex exhibit delta opioid receptor-mediated LTD (dOP-LTD), but are unaffected by MOR activation. Conversely, thalamostriatal glutamate release was strongly, but transiently, inhibited by MOR activation, but insensitive to activation of the delta receptor[2]. We therefore hypothesized that mOP-LTD is expressed

at other cortical inputs to DS. In addition, EtOH effects on input and region specificity of MOR plasticity have yet to be addressed.

In the current study, we use a combination of mouse brain slice electrophysiology, in vivo EtOH exposure, optogenetics, and conditional MOR knockout mice to probe MOR-mediated synaptic plasticity at the three sources of glutamatergic input to MSNs (cortical, thalamic, and CIN inputs) in the DS. Using these tools, we demonstrate that in vivo EtOH exposure persistently and specifically interferes with corticostriatal mOP-LTD exclusively within the DLS. Furthermore, we find that thalamostriatal MOR-mediated short-term depression and a novel form of plasticity, mOP-LTD of glutamate release from CINs, are unaffected by EtOH exposure. Finally, we identify the anterior insular cortex as the specific cortical input region that expresses alcohol-sensitive mOP-LTD in the DLS.

## Results

**MOR activation produces excitatory LTD in the DS.** To confirm previously established static mOP-LTD in the DS[2], electrically evoked excitatory postsynaptic currents (eEPSCs) were recorded before (10 min) and after a 5 min application of the MOR agonist [D-Ala[2], NMe-Phe[4], Gly-ol[5]]-enkephalin (DAMGO, 0.3 μM). Consistent with our previous observations[2], this brief application of DAMGO produced a persistent decrease in eEPSC amplitude in dorsal striatal MSNs (Fig. 1,

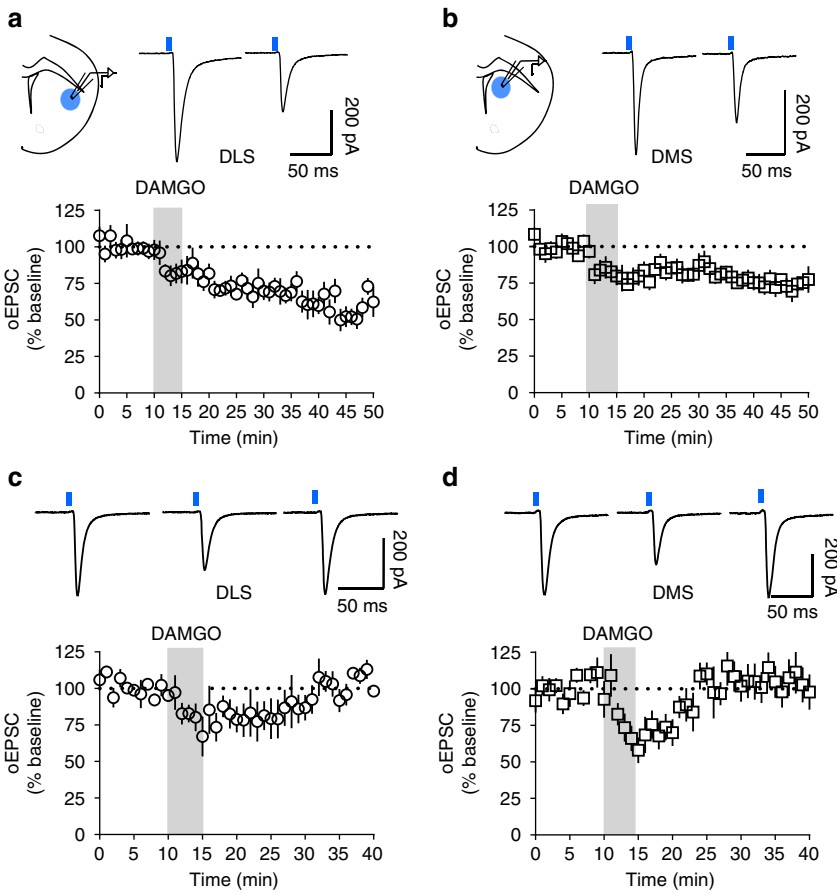

**Fig. 2** mOP-LTD occurs at corticostriatal but not thalamostriatal inputs. **a** Schematic of coronal brain slice showing the recording of oEPSCs by wide-field light stimulation (470 nm) in DLS of Ai32-Emx1Cre+ mice. mOP-LTD of cortical inputs occurs after the application of DAMGO (0.3 μM, 5 min) ($n = 8$ from 7 mice). **b** Schematic of coronal brain slice showing the recording of oEPSCs in DMS. mOP-LTD occurs in cortical inputs to DMS MSNs ($n = 9$ from 5 mice). **c, d** Activation of MORs (DAMGO 0.3 μM, 5 min) at thalamic inputs in Ai32-Vglut2Cre+ mice produces mOP-STD in the DLS ($n = 7$ from 3 mice) and the DMS ($n = 6$ from 2 mice). Data represent mean ± SEM. Traces in **a**, **b** represent average baseline oEPSCs and post-DAMGO oEPSCs. Traces in **c**, **d** represent average baseline oEPSC, average peak DAMGO inhibition oEPSC (14–19 min), and final 10 min after DAMGO average oEPSC

Supplementary Fig. 1a, b). This LTD is stable (≥40 min post DAMGO) and observed in both DLS (average EPSC amplitude for final 10 min of recording relative to 10 min baseline average: 74.5 ± 3.3%; Fig. 1b, c) and DMS (88.8 ± 3.2%; Fig. 1f, g). The observed glutamatergic depression was not a result of an increase in series resistance (Fig. 1d, h). These data demonstrate the presence of LTD mediated by the activation of striatal MORs in mouse DMS and DLS.

**mOP-LTD occurs at cortico but not thalamostriatal synapses.** To probe the glutamatergic circuits involved in mOP-LTD within the DS, we used transgenic mice and optogenetic tools to selectively activate cortical (Ai32-Emx1Cre+ mice) and thalamic (Ai32-Vglut2Cre+ mice) inputs onto dorsal striatal MSNs[28–31]. Interestingly, we found that when performing optical stimulation of cortical inputs, DAMGO induced robust mOP-LTD of optically evoked EPSCs (oEPSCs) in the DLS (59.1 ± 5.2%; Fig. 2a, Supplementary Fig. 1c) and a weaker mOP-LTD in the DMS (73.9 ± 3.3%; Fig. 2b, Supplementary Fig. 1d) of Ai32-Emx1Cre+ mice. However, using thalamic stimulation, DAMGO produced a labile plasticity at glutamatergic synapses that we term MOR-mediated short-term depression (mOP-STD). mOP-STD reached maximal suppression shortly after the end of DAMGO application (DLS: 79.5 ± 9.1%, Fig. 2c; DMS: 69.0 ± 7.7%, Fig. 2d) that returned to baseline levels in the DLS (100.6 ± 9.6%; Fig. 2c, Supplementary Fig. 1e) and DMS (104.9 ± 9.9%; Fig. 2d, Supplementary Fig. 1f). Thus, MOR activation at cortical afferent terminals is sufficient to induce glutamatergic LTD. Conversely,

MORs at thalamostriatal synapses are responsible for glutamatergic STD. These data demonstrate synapse specificity of MOR effects on neuroplasticity within the DS.

**In vivo exposure to EtOH reduces mOP-LTD only in DLS.** We tested whether in vivo administration of EtOH (2.0 g/kg, intraperitoneal (i.p.)) influenced the expression of mOP-LTD in the DS. Mice injected with saline (i.p.) 24 h before being killed showed normal LTD in both the DLS (76.1 ± 1.8%; Fig. 3b, c) and DMS (80.6 ± 3.0%; Fig. 3e, f) following bath application of DAMGO. In mice injected with EtOH 24 h before harvesting tissue, DAMGO-induced mOP-LTD was blunted in the DLS (90.2 ± 3.4%; Fig. 3a–c). However, EtOH pre-exposure did not influence mOP-LTD in the DMS (82.9 ± 2.3%; Fig. 3d–f). In sum, the data indicate that a single in vivo EtOH exposure is able to disrupt mOP-LTD, but this effect is regionally specific to the DLS.

**EtOH drinking disrupts mOP-LTD for several days in DLS.** Given that contingent and non-contingent alcohol exposure has been observed to differentially affect neurophysiology and neurochemistry in some cases[32–34], we compared the effects of the single exposure of experimenter-administered EtOH on mOP-LTD to voluntary binge drinking. Using the 'drinking-in-the-dark' (DID) mouse binge-like drinking paradigm[35], we compared the effects of DAMGO treatment on eEPSCs in mice that had consumed EtOH with those that had consumed sucrose via DID (Fig. 4a–c). Given that sugar also activates neural reward mechanisms and circuitry[36], sucrose consumption was used as a

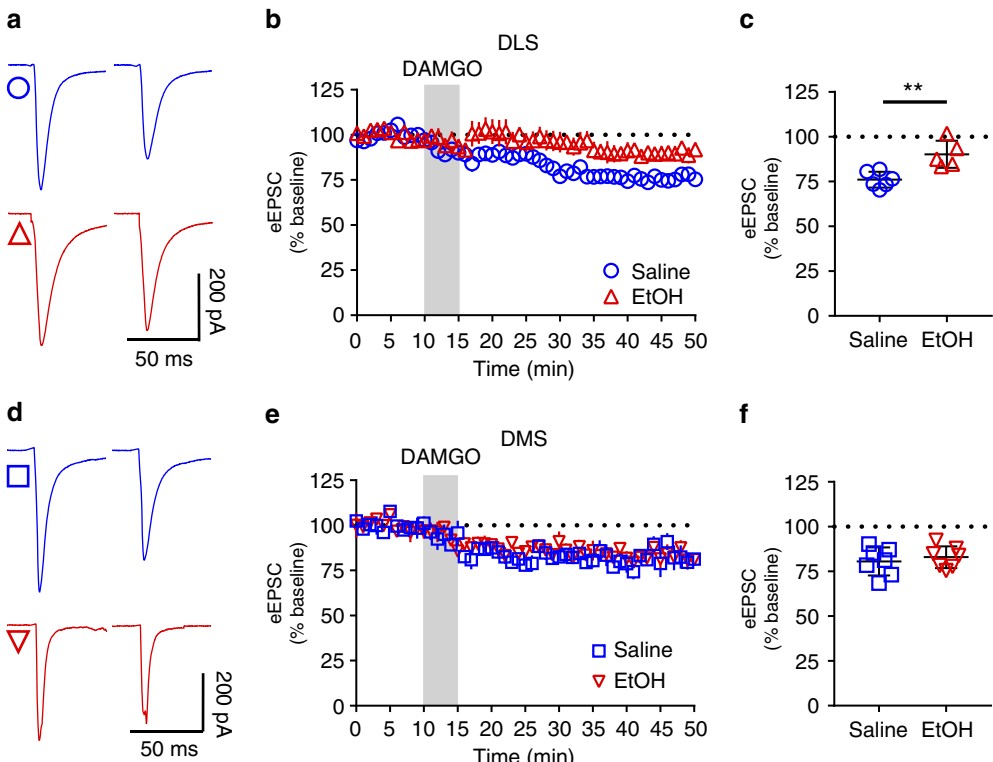

**Fig. 3** A single in vivo exposure to EtOH prevents induction of mOP-LTD specifically in DLS. C57BL/6J mice were injected (intraperitoneal) with saline or EtOH (2 g/kg). At 24 h after this injection, eEPSCs in MSNs from the DLS and the DMS were recorded. **a** Representative electrically evoked synaptic traces from the DLS before and after DAMGO (0.3 μM, 5 min) application in saline (blue traces) and EtOH (red traces)-injected mice. **b, c** EtOH disrupted mOP-LTD induced by DAMGO (0.3 μM, 5 min) in the DLS (P = 0.0038, $t_9$ = 3.87, saline n = 6 from 3 mice and EtOH n = 5 from 2 mice). **d** Representative electrically evoked synaptic traces from the DMS before and after DAMGO (0.3 μM, 5 min) application in saline (blue traces) and EtOH (red traces)-injected mice. **e, f** EtOH does not affect mOP-LTD induced by DAMGO (0.3 μM, 5 min) in the DMS (P = 0.54, $t_{12}$ = 0.627, saline n = 7 from 3 mice and EtOH n = 7 from 3 mice). Unpaired Student's t test. **P < 0.01. Further statistical analysis shows a significant interaction between exposure to in vivo EtOH and subregion (P = 0.038, F(1, 21) = 4.9, two-way ANOVA). Data represent mean ± SEM

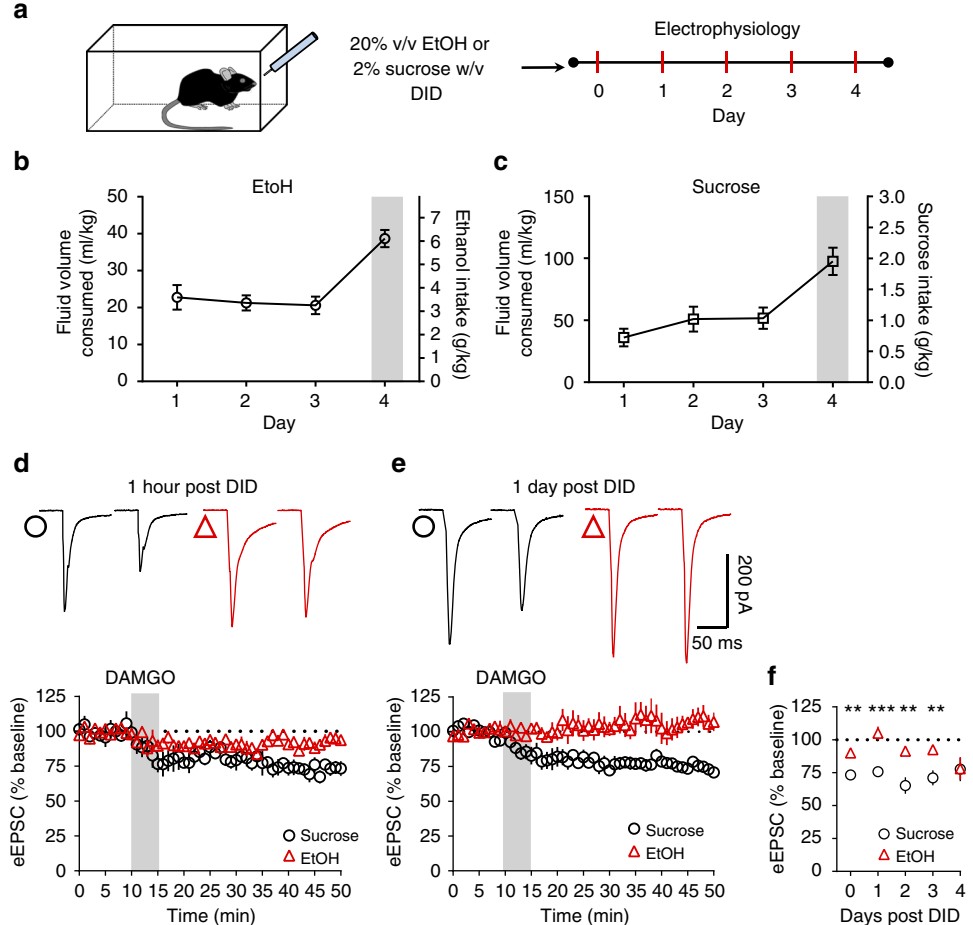

**Fig. 4** Voluntary binge EtOH drinking disrupts DLS mOP-LTD for several days. **a** Schematic figure summarizes the drinking in the dark (DID) paradigm. **b**, **c** Graph showing DID consumption data (EtOH and sucrose intake) in C57BL/6J mice ($n = 12$ EtOH and $n = 8$ sucrose). **d** EtOH consumption prevented mOP-LTD (0.3 μM DAMGO, 5 min) at 1 h post binge drinking, while sucrose mice showed normal mOP-LTD ($P = 0.0022$, $t_{12} = 3.873$; EtOH: $n = 8$ from 3 mice; Sucrose: $n = 6$ from 2 mice). **e** After 24 h of ending the DID procedure, EtOH mice show a lack of mOP-LTD in the DLS, but activation of MORs in sucrose mice produces mOP-LTD ($P = 0.0005$, $t_{10} = 5.101$; EtOH: $n = 6$ from 2 mice; sucrose: $n = 6$ from 2 mice). **f** EtOH disrupts mOP-LTD from **d** 1 h as well as **e** 1 day, 2 days ($P = 0.0014$, $t_{13} = 4.05$; EtOH: $n = 9$ from 3 mice; sucrose: $n = 8$ from 2 mice), and 3 days ($P = 0.0024$, $t_{15} = 3.646$; EtOH: $n = 9$ from 3 mice; sucrose: $n = 8$ from 2 mice) after the cessation of DID drinking. At day 4 post DID, the activation of MORs by DAMGO in both EtOH and sucrose mice induces equivalent mOP-LTD ($P = 0.974$, $t_6 = 0.0342$; EtOH: $n = 4$ from 1 mouse; Sucrose: $n = 4$ from 2 mice). Gray bars highlight the 4 h DID session on day 4. Unpaired Student's $t$ test. **$P < 0.01$, ***$P < 0.01$. Data represent mean ± SEM

natural reward control. A parallel cohort of mice was used to estimate blood ethanol concentrations (BECs) to avoid stress effects of blood sampling on mOP-LTD. Blood samples were not collected from mice to be used in electrophysiology experiments due to the concern that stress significantly influences MSN synaptic plasticity measurements[37]. EtOH consumption in mice used in electrophysiology experiments was not different from this parallel cohort (Supplementary Fig. 2a, b). EtOH consumption was significantly predictive of BECs as determined by a regression analysis (Supplementary Fig. 2c, d). Using the obtained best-fit regression line, we predicted that the BECs of mice used in electrophysiology experiments were likely between ~80 and 150 mg/dl on day 4 of DID given their measured EtOH intake (Supplementary Fig. 2c, d). Altogether, we are confident that the mice used in electrophysiology plasticity experiments consumed physiologically relevant, binge-like levels of EtOH. Electrophysiology experiments evaluated the effects of sucrose or EtOH binge drinking on mOP-LTD in the DLS at increasing intervals following the last drinking session. Given that EtOH had no effect on DMS mOP-LTD (Fig. 3d–f), we focused the remainder of our experiments on the DLS. We found that MSNs from sucrose-

drinking mice exhibited normal mOP-LTD 1 h following the last DID session ($73.3 \pm 3.6\%$; Fig. 4d, f). mOP-LTD was also apparent at 1 ($75.8 \pm 2.8\%$; Fig. 4e, f), 2 ($65.3 \pm 6.3\%$: Fig. 4f), 3 ($71.1 \pm 5.8\%$; Fig. 4f), and 4 ($77.7 \pm 9.1\%$) days following the final sucrose DID session. mOP-LTD in MSNs from EtOH-drinking mice was disrupted 1 h after the final day 4 DID session ($90.1 \pm 2.6\%$; Fig. 4d, f). Intriguingly, this disruption persisted for 1 ($105.1 \pm 5.0\%$; Fig. 4e,f), 2 ($91.4 \pm 3.2\%$; Fig. 4f), and 3 ($92.6 \pm 2.2\%$; Fig. 4f) days after the cessation of DID drinking. By day 4 however, mOP-LTD recovered to normal levels ($78.1 \pm 8.5\%$; Fig. 4f). These results suggest that a single experimenter-administered EtOH exposure and repeated binge-like EtOH drinking can similarly disrupt MOR plasticity in the DLS. Given that intake of the natural reward sucrose had no effect on mOP-LTD, these observations cannot simply be due to the activation of reward mechanisms. These observations are therefore EtOH specific.

**Effect of EtOH is specific to corticostriatal synapses**. To further characterize the synapse-specific effects of EtOH on MOR-mediated plasticity in DLS, we probed the effect of a single in vivo

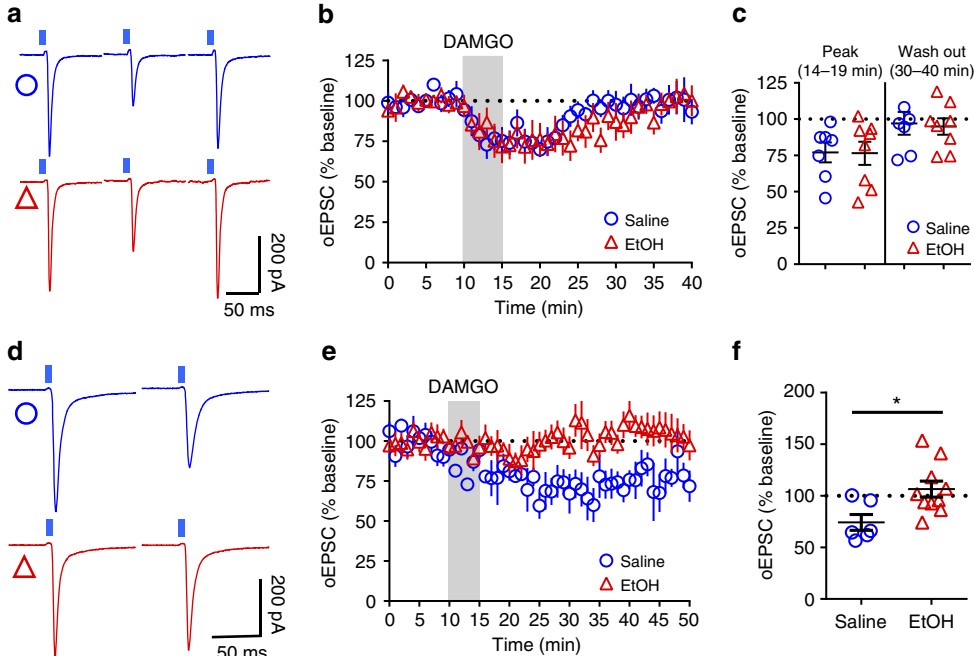

**Fig. 5** A single in vivo exposure to ethanol disrupts mOP-LTD at cortical inputs but not mOP-STD at thalamic inputs. **a** Representative light-evoked synaptic traces (oEPSC) from the DLS during baseline, immediately after DAMGO (0.3 μM, 5 min) application (14–19 min), and final 10 min of recording in saline (blue traces) and EtOH (red traces)-injected Ai32-Vglut2Cre+ mice. **b, c** EtOH does not affect peak inhibition ($P = 0.96$, $t_{13} = 0.05$, saline $n = 7$ from 2 mice and EtOH $n = 8$ from 2 mice) or mOP-STD induced by DAMGO in the DLS ($P = 0.824$, $t_{13} = 0.227$, saline $n = 7$ from 2 mice and EtOH $n = 8$ from 2 mice). **d** Representative oEPSC traces from DLS before and after DAMGO (0.3 μM, 5 min) application in saline (blue traces) and EtOH (red traces)-injected Ai32-Emx1Cre+ mice. **e, f** EtOH produces a disruption of mOP-LTD from cortical inputs induced by DAMGO (0.3 μM, 5 min) in the DLS ($P = 0.016$, $t_{14} = 2.75$, saline $n = 6$ from 2 mice and EtOH $n = 10$ from 4 mice). Unpaired Student's t test. *$P < 0.05$. Data represent mean ± SEM

EtOH exposure in Ai32-Emx1Cre+ and Ai32-Vglut2Cre+ mice. We found that mOP-STD of thalamic inputs (Fig. 5a–c) is unaffected by EtOH exposure at the peak inhibition (77.0 ± 6.9% saline vs 76.5 ± 7.9% EtOH; Fig. 5c) and at 40–50 min (97.1 ± 7.9% saline vs 94.9 ± 5.7% EtOH; Fig. 5c). However, activation of MORs on cortical inputs produced a stable LTD in saline-injected mice (74.2 ± 7.7%; Fig. 5d–f) and a single EtOH injection was able to disrupt this mOP-LTD 24 h later (106.5 ± 7.8%; Fig. 5d–f). Finally, neither bath application of 20 (91.62 ± 4.0 %) or 40 mM EtOH (93.5 ± 4.3 %) induced a synaptic depression of oEPSCs nor blocked mOP-LTD in the DLS (20 mM: 66.4 ± 7.8 % and 40 mM: 67.7 ± 6.1 %; Supplementary Fig. 3). Furthermore, in vivo EtOH exposure did not affect the stimulus intensity response of eEPSCs (Supplementary Fig 4a, b), synaptic properties of spontaneous EPSCs (sEPSCs: Supplementary Fig. 5), or miniature EPSCs (mEPSCs: Supplementary. 6) in the DLS. These observations indicate that EtOH does not appear to induce LTD on its own and thus occludes ex vivo probing of mOP-LTD.

CINs are known to gate corticostriatal glutamatergic signaling[38–40] and are also reported to co-release glutamate[9]. Furthermore, it has been shown that MORs regulate CIN activity[41,42]. It is therefore possible that mOP-LTD could also occur via MOR regulation of CIN activity. To test this, we expressed Channelrhodopsin2 (ChR2) in dorsal striatal CINs in ChATCre+ mice (Fig. 6a, b)[43,44]. We found that application of DAMGO produced a persistent reduction in the magnitude of oEPSCs recorded in MSNs produced by photostimulation of CINs in the DLS of saline-injected mice (76.8 ± 4.2%). This novel form of LTD, CIN-mOP-LTD, was equivalent in EtOH-injected ChR2-expressing ChATCre+ mice (81.5 ± 3.8%; Fig. 6d, e). To further characterize the glutamatergic CIN activity, we blocked

glutamate currents using the AMPA receptor antagonist, NBQX (5 μM). NBQX was incapable of blocking the evoked current entirely, suggesting that the remaining current (34.9 ± 10.5 %) is likely due to nAChR activation (Supplementary Fig. 7a-c). Interestingly, the application of the α7 antagonist methyllycaconitine (MLA: 100 nM) did not affect the induction of mOP-LTD by DAMGO (Supplementary Fig. 7d-f). Given that α7 nAChRs selectively affect glutamate release from presynaptic corticostriatal terminals[40,45,46], these negative results suggest that CIN-mOP-LTD measured in DLS MSNs is not a result of an indirect effect of acetylcholine acting on MOR-expressing cortical terminals, but rather occurs directly via a reduction in glutamate release from the CINs onto MSNs. Finally, we used MOR-flox/ChATCre+ mice injected with AAV9.DIO.ChR2 in DLS (Supplementary Fig. 8a) to confirm that MORs on CINs, themselves, are responsible for this novel form of mOP-LTD. MOR-flox/ChATCre+ mice did not express CIN-mOP-LTD (97.9 ± 5.7 %; Supplementary Fig. 8b-d). These data, along with our cortical and thalamic input data, indicate that EtOH has a synapse-specific effect rather than a MOR plasticity type-specific effect.

**mOP-LTD specifically occurs at insular cortex inputs to DLS.** Our data to this point indicate that MOR activation is sufficient to induce EtOH-sensitive mOP-LTD at corticostriatal synapses. To further establish the necessary role of MORs on cortical terminals in mOP-LTD, we used conditional MOR knockout mice (MOR-flox/Emx1Cre+), lacking MORs exclusively on cortical inputs (Fig. 7a, b). DAMGO application was incapable of producing mOP-LTD of eEPSCs in MOR-flox/Emx1Cre+ mice (97.4 ± 3.1%; Fig. 7c) compared to wild-type MOR-flox/Emx1cre-

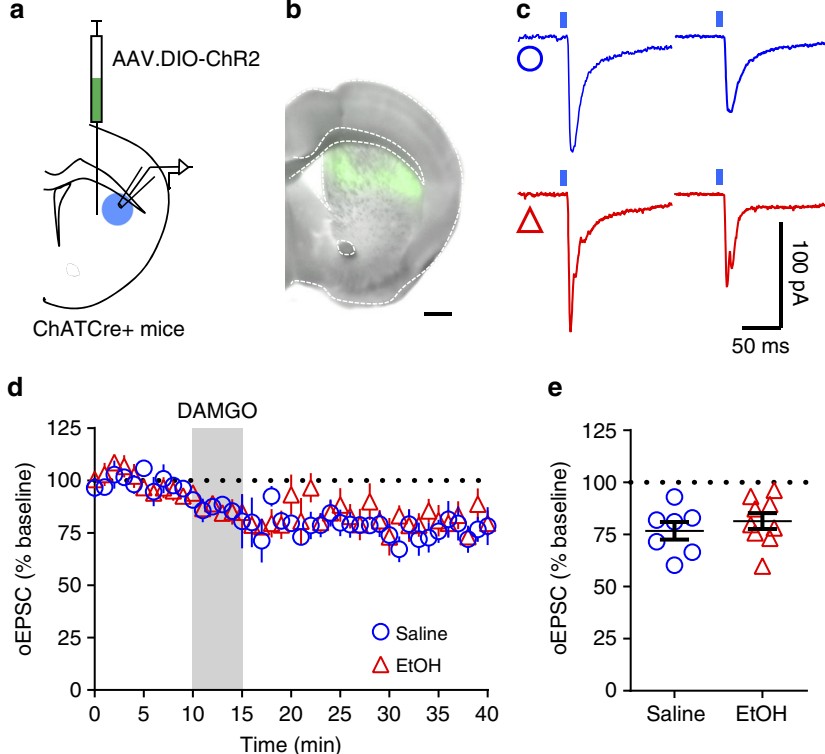

**Fig. 6** mOP-LTD of glutamate release at cholinergic interneuron synapses is not affected by EtOH. **a** Schematic figure of the injection paradigm enabling optogenetic recording from ChATCre+ DLS MSNs. A cre-recombinase-dependent AAV vector coding for ChR2 (AAV.DIO.ChR2) was injected 14 days prior to saline and EtOH treatment. **b** Coronal brain slice showing the infection of cholinergic interneurons in the DS (bar scale = 500 μm). **c** Representative light-evoked synaptic traces from the DLS before and after DAMGO (0.3 μM, 5 min) application in saline (blue traces) and EtOH (red traces)-injected mice. **d**, **e** EtOH was unable to disrupt mOP-LTD at cholinergic interneuron synapses in the DLS ($P = 0.42$, $t_{14} = 0.83$, saline $n = 7$ from 2 mice and EtOH $n = 9$ from 4 mice). Unpaired Student's $t$ test. Data represent mean ± SEM

counterparts (76.0 ± 4.4%; Fig. 7c). These data indicate that MORs on cortical inputs are both necessary and sufficient to induce mOP-LTD of glutamatergic transmission within the DLS.

Having characterized and established the role of MORs on corticostriatal inputs, we further attempted to determine the specific cortical region involved in mOP-LTD. We infused AAV-ChR2 vector into medial prefrontal (mPFC) and orbitofrontal cortex (OFC) to evaluate the role MORs on those cortical inputs. We found that glutamate release from both mPFC and OFC inputs to DLS was unaffected by the activation of MORs (Supplementary Fig. 9). Our prior work also determined that motor cortex was not involved in the generation of mOP-LTD[2]. It is therefore possible that mOP-LTD could occur in a non-classical addiction-related cortical-DLS circuit. Another cortical area with substantial input to DS is the anterior agranular insular cortex[11,47]. We hypothesized a possible role of anterior insular inputs to DLS MSNs in regulating mOP-LTD. To test this, we expressed ChR2 in anterior insular cortex and recorded oEPSCs in MSNs in the DLS (Fig. 8a). We noted that anterior insular projections were mainly found in the DLS and nucleus accumbens in our brain slices (Fig. 8a). DAMGO treatment was indeed sufficient to produce glutamatergic mOP-LTD of optically stimulated anterior insular corticostriatal synapses (76.7 ± 5.2 %, Fig. 8b, c). Furthermore, application of the MOR antagonist CTAP (D-Phe-Cys-Tyr-D-Trp-Arg-Thr-Pen-Thr-NH₂) completely blocked the induction of mOP-LTD at these synapses (96.6 ± 5.8 %; Fig. 8c, d). These data suggest that MORs at anterior insular cortex inputs to the DLS are capable of expressing mOP-LTD in the DLS. Next, using i.p. EtOH injection (2.0 g/kg), we evaluated whether anterior insular inputs are sensitive to in vivo

EtOH exposure. We found that EtOH was able to disrupt mOP-LTD from anterior insular inputs 24 h post injection (72.9 ± 5.8% saline vs 102.4 ± 5.0 % EtOH; Fig. 8e-g), similar to our earlier observations reported in Figs. 3–5.

To further characterize this new role of anterior insular cortex dorsal striatal inputs, we used MOR-flox mice injected with AAV-cre vector to knock out the expression of MORs specifically in the anterior insular cortex (Fig. 9a). Using electrical stimulation and recording in the DLS (Fig. 9a), we found that DAMGO application did not produce mOP-LTD (93.6 ± 5.8%, Fig. 9b), although a subsequent high-frequency stimulation (HFS) train was able to induce eCB-LTD after application of DAMGO (DAMGO: 89.1 ± 5.0%; HFS: 56.6 ± 10.2%, Fig. 9c, d). Finally, MOR knockout from the anterior insular cortex leaves dOP-LTD intact (Supplementary Fig. 10). Altogether, these data suggest that MORs specifically on anterior insular cortex inputs are both necessary and sufficient to induce mOP-LTD in the DLS and are uniquely affected by in vivo ethanol exposure.

## Discussion

The current study confirmed previously established mOP-LTD in DLS[2] (Fig. 1) and now demonstrates that it occurs in mouse DMS as well (Fig. 1). MOR-mediated inhibition of glutamate release could occur at three distinct synapse types in the DS: the glutamatergic inputs from cortex or thalamus, or driven by CIN activation[9]. Using Cre-dependent ChR2-expressing transgenic mice[28,29], we determined that corticostriatal glutamatergic inputs are involved in mOP-LTD, while thalamostriatal synapses expressed mOP-STD (Fig. 2). Indeed, ablating MOR expression

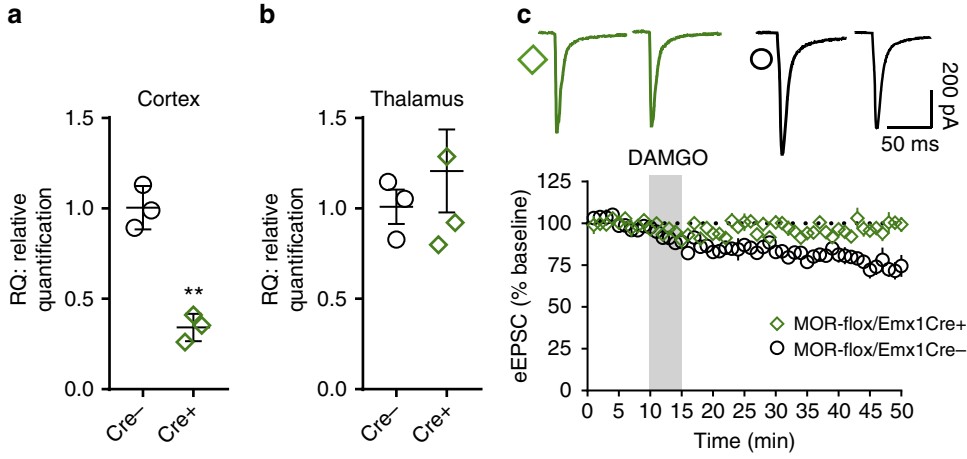

**Fig. 7** MORs from cortical inputs are necessary to induce mOP-LTD in the DLS. **a** Quantitative PCR analysis showing that the deletion of MOR in the frontal cortex of MOR-flox/Emx1Cre+ mice reduces MOR mRNA expression (Cre−: $1.01 \pm 0.070$; Cre+: $0.34 \pm 0.044$; $P = 0.0013$, $t_4 = 8.113$). Measurable MOR signal in Cre+ mice likely reflects expression in interneurons. **b** Quantitative PCR showing intact MOR expression in the thalamus of MOR-flox/Emx1Cre+ mice (Cre−: $1.01 \pm 0.10$; Cre+: $1.21 \pm 0.23$; $P = 0.5151$, $t_5 = 0.7002$). **c** The elimination of MORs from cortical afferents in MOR-flox/Emx1Cre mice prevented mOP-LTD in DLS ($P = 0.0025$, $t_{10} = 4.006$, $n = 6$ each, from 2 MOR-flox/Emx1Cre− mice and 3 MOR-flox/Emx1Cre+ mice). RQ relative quantification. Unpaired Student's $t$ test. **$P < 0.01$. Data represent mean ± SEM

in corticostriatal synapses leads to a selective disruption of mOP-LTD using electrical stimulation, suggesting that presynaptic MORs are mediating this plasticity in the DS (Fig. 7). In addition, we presented a novel characterization of mOP-LTD of direct CIN glutamate release within the DLS (Fig. 6, Supplementary Fig. 7). Activation of MORs located on CINs appear to be responsible for this form of LTD (Supplementary Fig. 8).

Until now, the cortical region(s) that mediate mOP-LTD were unknown. The DS receives major projections from many cortical regions including, but not limited to, mPFC, OFC, and sensorimotor areas. Here we used optogenetic stimulation of mPFC and OFC inputs to the DLS and established that presynaptic MOR activation from those cortical terminals did not produce glutamatergic synaptic depression. Furthermore, our previous findings indicate that motor cortex is also not involved in the induction of mOP-LTD[2]. The lack of mOP-LTD, or even an acute response to DAMGO, from these regions can be explained as a lack of MOR availability to induce synaptic depression. Therefore, we related our loss of mOP-LTD in MOR-flox/Emx1Cre+ mice (Fig. 7) to another cortical region. The specific optical stimulation of anterior insular terminals and activation of presynaptic MORs leads to synaptic plasticity that was blocked by the MOR antagonist CTAP. In addition, the knockout of MORs from anterior insular inputs completely abolished mOP-LTD in DLS, but not eCB-LTD or dOP-LTD, indicating the specificity of this circuit for MOR-mediated regulation of glutamate release. Our results demonstrate that the anterior insular cortex-to-DLS input is the critical projection that expresses MOR-mediated long-term plasticity. Alterations in the plasticity of this input to DLS suggest that this circuit could mediate behaviors such as decision making[48], interoception[49], pain[50,51], taste learning behavior[52,53], compulsive alcohol use[54], cocaine seeking and intake[55,56], and opiate-related positive and negative affective learning[57]. Thus, now that we know the specific cortical input that is responsible for mOP-LTD in the DLS, we can begin new efforts to interpret the behavioral ramifications of our EtOH-mediated mOP-LTD disruption.

EtOH exposure-mediated disruption of mOP-LTD is consistent with what we have previously shown for oxycodone exposure[2]. Our findings show that both experimenter-administered and voluntary binge-like EtOH exposure can ablate long-term glutamatergic synaptic depression mediated by the activation of MORs. Furthermore, the effect of voluntary

binge-like drinking on mOP-LTD persisted for 3 days following the cessation of EtOH access. This EtOH-disrupted mOP-LTD was similar to previous observations with i.p. oxycodone treatment, an effect which also persisted for multiple days post treatment[2]. Although mOP-LTD was not assessed 1 h post EtOH injection in this study, we expect that mOP-LTD would be similarly disrupted as was observed in the drinking experiment. Our previous data show that i.p. oxycodone blocks the induction of mOP-LTD 1 h after exposure[2] and persists for 3 days. This is very similar to the findings of the binge drinking experiment in the current study. It is of great interest that the effects of EtOH and oxycodone both persisted long after even limited exposure had ceased. Future work will need to determine if a longer history of EtOH drinking will produce more persistent effects (i.e., beyond 3 days) on plasticity as others have shown for cocaine[58,59].

Both oxycodone and EtOH exposure suppress eCB-LTD in the DS, plasticity that also interacts with mOP-LTD[2,13,15,17]. CINs are important components of corticostriatal eCB-LTD[38], and EtOH effects on dorsal striatal output have been proposed to operate by affecting cholinergic signaling[60]. Therefore, considering the role of MORs in modulating CIN activity[41,42], it was important for us to address the possibility that the effects of EtOH on mOP-LTD could be occurring as a result of altered CIN-driven glutamate signaling. However, we found that our observed EtOH effects were not related to altered CIN-mOP-LTD (Fig. 6). These data further support that EtOH has synapse-specific effects, rather than just plasticity type-specific effects. mOP-LTD in the DMS and of CIN-driven glutamate release in DLS are both unaffected by EtOH treatment, whereas mOP-LTD at corticostriatal DLS synapses is selectively disrupted. Specifically, having demonstrated that the activation of anterior insular inputs onto the DLS MSNs are necessary and sufficient to induce mOP-LTD, the finding that EtOH exposure disrupts the MOR-mediated LTD occurring at this projection suggests that insular MORs may be important targets of adaptation in EtOH-related behaviors. Previous studies have demonstrated a role for MORs in mediating responding for natural rewards[18], but our data here demonstrate that sucrose has no effect on mOP-LTD (Fig. 4). Thus, insular cortex-DLS mOP-LTD may therefore be particularly sensitive to drugs of abuse.

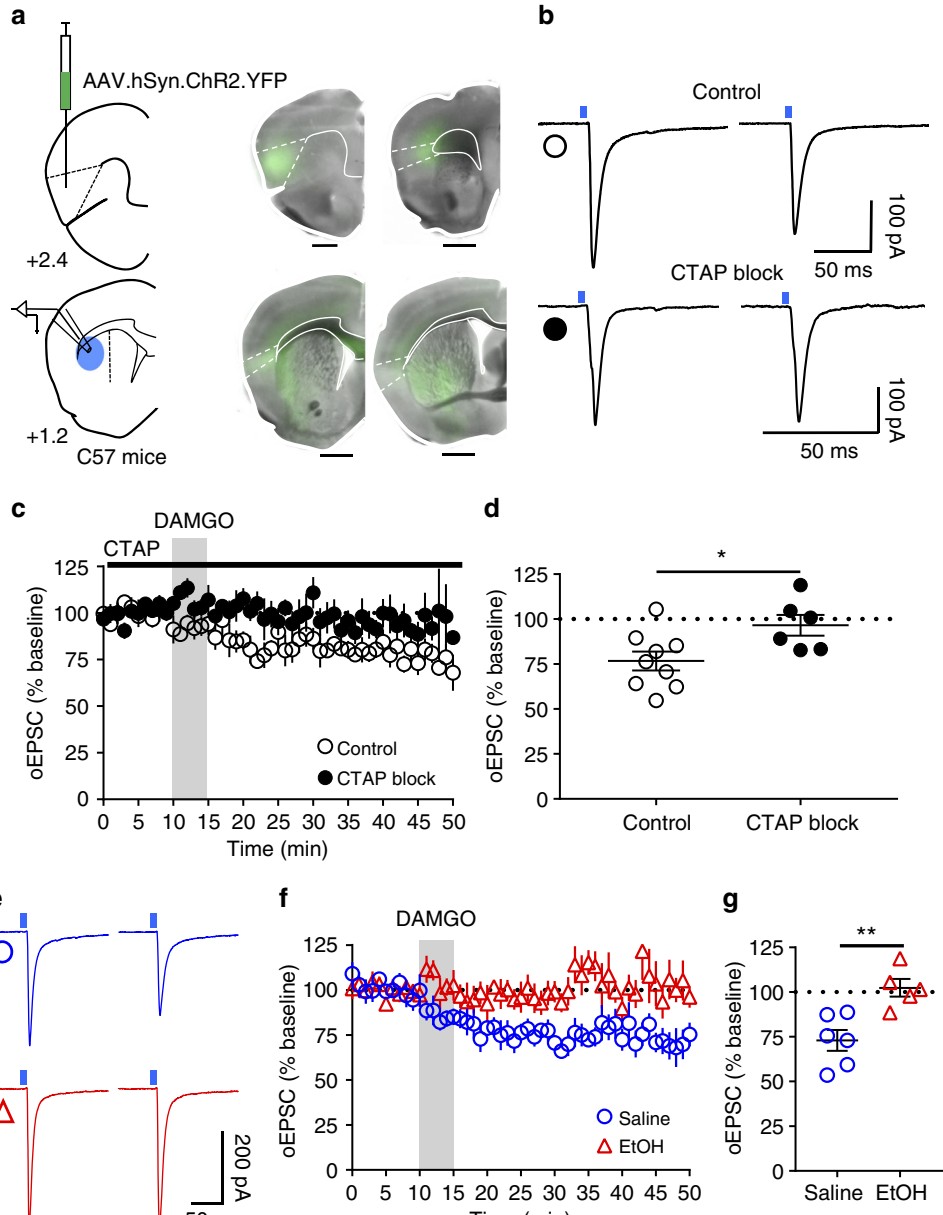

**Fig. 8** mOP-LTD occurs at anterior insular cortex inputs to DLS and is ethanol sensitive. **a** Schematic figure of the injection paradigm enabling optogenetic recording from C57BL/6J DLS MSNs. An AAV vector coding for ChR2 (AAV.hSyn.ChR2) was injected 14 days prior to recordings. Coronal brain slices showing the infection of anterior insular cortex sending projections to the DLS and nucleus accumbens. Bar scales: +2.4, 200 μm; +1.7, 1000 μm; +1.2, 1000 μm; +0.1, 1000 μm. **b** Representative light-evoked synaptic traces from the DLS before and after DAMGO (0.3 μM, 5 min) and blocked with MOR antagonist CTAP (1 μM). **c**, **d** DAMGO application induced mOP-LTD at anterior insular terminals in the DLS and this MOR-mediated LTD was blocked by the application of CTAP ($P = 0.027$, $t_{13} = 2.49$, Control: $n = 9$ from 3 mice; CTAP: $n = 6$ from 2 mice). **e** Representative oEPSC traces from the DLS before and after DAMGO (0.3 μM, 5 min) application in saline (blue traces) and EtOH (red traces)-injected C57BL/6J mice. **f**, **g** EtOH exposure produced a disruption of mOP-LTD from anterior insular inputs induced by DAMGO (0.3 μM, 5 min) in the DLS ($P = 0.0046$, $t_9 = 3.75$; Saline: $n = 6$ from 2 mice; EtOH: $n = 5$ from 3 mice). Unpaired Student's t test. *$P < 0.05$, **$P < 0.01$. Data represent mean ± SEM

The selective effects of EtOH on dorsal striatal subregion (DLS, not DMS), type of plasticity (LTD, not STD), and synapse type (insular corticostriatal, not thalamostriatal or cholinergic) provide some interesting possibilities for how EtOH may influence behavior in that it spares opioid signaling at some dorsal striatal synapses, while weakening it at others. Specific EtOH effects could be due to the different innervations that each dorsal striatal subregion receives[10] (insular inputs reach mainly DLS, Fig. 8a), resulting in different synaptic environments. As such, this may explain differences in their sensitivity to EtOH. The lack of a statistical difference between subregions in mOP-LTD magnitude

(Fig. 3) also suggests that no pre-existing differences in the relative strengths of mOP-LTD in DMS versus DLS could have precluded the potential to observe an EtOH effect in the DMS.

It has been proposed that addiction-related dorsal striatal neuroadaptations facilitate the development of habit learning in the context of substance-related behavior[61]. The specificity and duration of our observed EtOH effect (3 days post drinking) has significant implications for how acute EtOH exposure produces functional changes selectively within the DLS, a brain region known to promote habitual EtOH seeking[14]. Opioid signaling has also been implicated in habit learning[62] and is clearly involved in

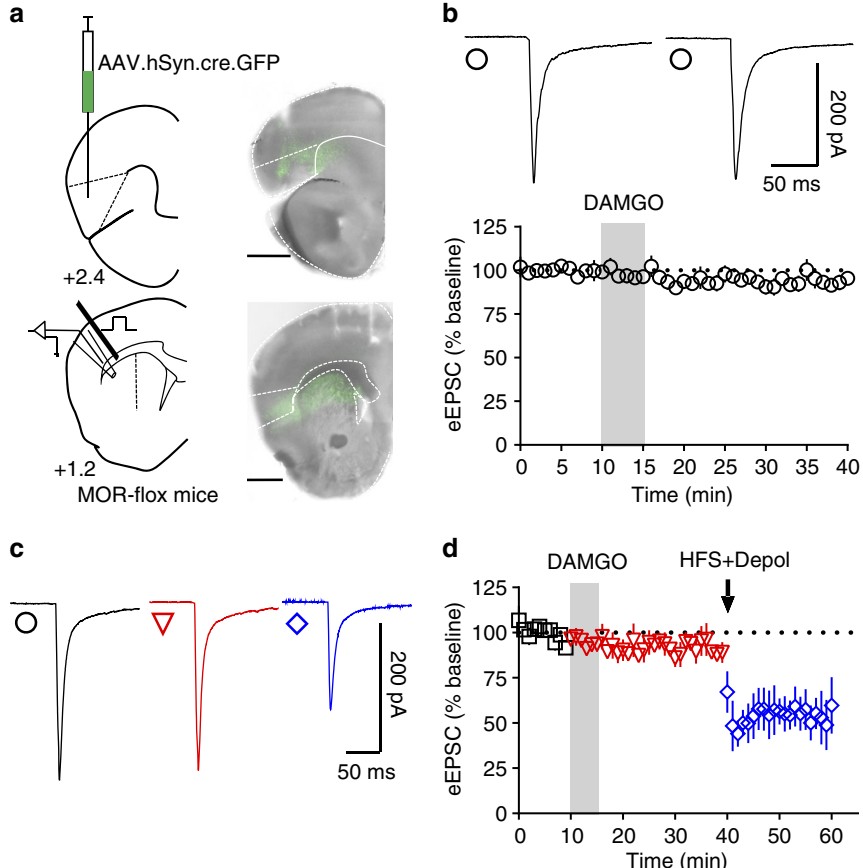

**Fig. 9** Anterior insular MORs are necessary to induce mOP-LTD in the DLS. **a** Schematic figure of coronal brain slice showing the recording of EPSCs evoked by focal electric stimulation in the DLS of MOR-flox mice. An AAV vector encoding for cre-recombinase (AAV.hSyn.cre) was injected 8 weeks prior to recordings. Coronal brain slice showing the infection of anterior insular cortex and dorsal striatal terminal expression (bar scale = 1000 μm). **b** Representative electrically evoked synaptic traces and a recording time course from the DLS before and after DAMGO (0.3 μM, 5 min) showing the loss of mOP-LTD mediated by anterior insular MORs ($P = 0.136$, $t_5 = 1.775$, $n = 6$ from 4 mice). **c, d** Representative electrically evoked synaptic traces and a recording time course from the DLS before and after DAMGO (0.3 μM, 5 min) and high-frequency stimulation (HFS) coupled with depolarization (4 pulses of 100 Hz, 10 s inter-pulse interval). DAMGO application was incapable of producing mOP-LTD in the DLS of anterior insula MOR knockout mice. However, HFS induced a strong reduction of eEPSCs in the DLS, indicating intact eCB-LTD (baseline vs DAMGO: $P = 0.262$, $t_3 = 1.378$; baseline vs HFS: $P = 0.0082$, $t_3 = 6.259$; DAMGO vs HFS: $P = 0.0467$, $t_3 = 3.273$; $n = 4$ from 3 mice). Data analyzed with Student's paired $t$ tests. Data represent mean ± SEM

behavioral responding for alcohol[18]. Alternatively, data have also demonstrated that MOR activation in the DLS of rats significantly enhances the incentive salience of reward-associated cues[63]. This enhancement essentially manifests as the cue(s) itself serving as a primary reward. Individuals with substance use disorders may more strongly seek these cues over time, thus increasing craving and the likelihood of consumption[64,65]. The extent of such EtOH-induced functional adaptations in the DLS opioid signaling may therefore be important for an individual's risk for developing an AUD. The observation that MORs are important regulators of EtOH conditioned place preference[66] and locomotor sensitization[67], behaviors that develop over time with repeated exposure, suggests that these behavioral changes are likely accompanied by alterations in synaptic plasticity. During the pre-dependent stage of alcohol exposure, changes in the functional dynamics of pre-synaptic DLS MOR signaling could occur, perhaps reflecting a critical transition state to problematic alcohol use as has been demonstrated in opiate-related positive and negative affective learning governed by the insular cortex[57]. This hypothesis appears to be consistent with our finding that EtOH exposure interrupts the functional ability of the MOR to suppress gluta-matergic drive in the DLS. Others have utilized the DID model to determine if EtOH exposure produces changes in basal

glutamatergic signaling, but did not find any effect[68]. Our data support these findings as well (Supplementary Figs. 3-6). This suggests that EtOH is not preventing mOP-LTD expression in slice by occlusion, but rather is preventing its induction. Additional work is needed to determine the molecular mechanisms underlying this disruption. Identifying these mechanisms may allow for the discovery of pharmacological targets as intervention strategies for the excessive incentive salience and/or habitual component(s) of AUDs.

Finally, an important question for these experiments is whether stress influenced mOP-LTD measurements. Indeed, stressors can disrupt synaptic plasticity in MSNs[37]. As mice received i.p. injections or underwent surgery in some of our experiments, this is an important consideration. In addition, mice could have been in a dysphoric state of EtOH withdrawal in some experiments. In our case, we performed some experiments with saline-injected controls. Comparatively, mOP-LTD data from these animals are highly similar to naive animals (Figs. 1 and 3). In addition, the DMS data in Fig. 3 indicate that EtOH-injected mice show no effect of EtOH exposure on mOP-LTD. As this was 24 h later, any sort of withdrawal state that may interfere with plasticity would be present, but no effect was found. Furthermore, in our experiment with MOR-flox/Emx1Cre mice (Fig. 7c), animals

were not injected and mOP-LTD was blocked by the knockdown of MOR from the cortex in the absence of any stressors. We therefore believe that the possibility that stress could have been responsible for or contributed to any of our plasticity measurements is unlikely.

In conclusion, our data provide the first evidence that MORs produce different types of synaptic plasticity at distinct dorsal striatal glutamatergic synapses and that EtOH has persistent, yet selective, effects on this plasticity. We found that mOP-LTD in the DLS was specifically expressed at anterior insular cortex inputs and EtOH affects this plasticity in a unique manner. How mOP-LTD at insular synapses may be involved in DS-dependent behaviors and how this relates to EtOH effects on the expression of these behaviors will be of interest in our future investigations.

## Methods

**Animals and materials**. Animal care and experimental protocols for this study were approved by the Institutional Animal Care and Use Committee at the Indiana University School of Medicine and all guidelines for ethical protocols and care of experimental animals established by the NIH (National Institutes of Health, Maryland, USA) were followed. Male C57BL/6J mice were ordered from the Jackson Laboratory (Bar Harbor, Maine, USA). Ai32-Emx1Cre, Ai32-Vglut2Cre, and ChATCre transgenic mice were bred and genotyped in-house (Original stock strains: Ai32: JAX #012569; ChATCre: JAX #006410; Emx1Cre: JAX #005628; Vglut2Cre: JAX #016963). Conditional MOR knockout mice (MOR-flox) were a generous gift from Dr. Jennifer Whistler (UCSF)[69]. All transgenic mice used in these studies have been backcrossed to C57BL/6J mice for a minimum of 7 generations. MOR-flox mice were crossed with Emx1Cre mice to produce cortical projection neuron MOR knockout mice (MOR-flox/Emx1Cre) and crossed with ChATCre mice to produce MOR knockout from CINs. The mice used in these studies were between postnatal day (PND) 32 and 90 at the time of experimentation (with the exception of ChATCre mice that were ~PND 60–150 and MOR-flox AAV-cre-injected mice ~PND 116–136), with all mice involved in the drinking studies between PND 70 and 80. Animals were group-housed (with the exception of the voluntary drinking experiment outlined below) in a standard 12 h light/dark cycle (lights on at 0800 h). Humidity in the vivarium was held constant at 50% and food and water were available ad libitum, except where noted otherwise. Drugs and reagents were purchased from Fisher Scientific, Tocris, Sigma-Aldrich, or Bachem Americas.

**Brain slice preparation**. Immediately following killing via decapitation under deep isoflurane anesthesia, the brain was quickly excised and placed in an ice-cold cutting solution containing (in mM): 194 sucrose, 30 NaCl, 4.5 KCl, 1 MgCl$_2$, 26 NaHCO$_3$, 1.2 NaH$_2$PO$_4$, 10 Glucose saturated with a mixture of 95% O$_2$ and 5% CO$_2$, and sliced to a thickness of 280 μm on a vibratome (Leica VT1200S, Germany). Slices were transferred to an artificial cerebrospinal fluid (aCSF) solution containing (in mM): 124 NaCl, 4.5 KCl, 1 MgCl$_2$, 26 NaHCO$_3$, 1.2 NaH$_2$PO$_4$, 10 Glucose, 2 CaCl$_2$ (310–320 mOsm) saturated with 95% O$_2$/5% CO$_2$ at 30 °C for 1 h before being moved to room temperature. When ready for recording, slices were transferred to a recording chamber continuously perfused with aCSF solution saturated with 95% O$_2$/5% CO$_2$.

**Electrophysiology recordings**. Whole-cell, voltage-clamp recordings of EPSCs from MSNs were carried out at 29–32 C° and aCSF was continuously perfused at a rate of 1–2 ml/min. Whole-cell patch clamp recordings in the voltage-clamp configuration were made from MSNs using a Multiclamp 700B amplifier (Axon Instruments, Union City, CA). Slices were visualized on an Olympus BX51WI microscope (Olympus Corporation of America). MSNs were identified by their size, membrane resistance, and capacitance. Picrotoxin (50 μM) was added to the aCSF for recordings to isolate excitatory transmission. Patch pipettes were prepared from filament-containing borosilicate micropipettes (World Precision Instruments) using a P-1000 micropipette puller (Sutter Instruments, Novato, CA), having a 2.0–3.5 MΩ resistance. The internal solution contained (in mM) 120 CsMeSO$_3$, 5 NaCl, 10 TEA-Cl, 10 HEPES, 5 lidocaine bromide, 1.1 EGTA, 0.3 Na-GTP, and 4 Mg-ATP (pH 7.2 and 290–310 mOsm). MSNs were voltage clamped at −60 mV for the duration of the recordings. For DLS recordings, a tungsten bipolar stimulating electrode (PlasticsONE, Roanoke, VA) was placed at the border of the white matter of the external capsule. For DMS recordings, the stimulating electrode was placed at the border of the overlying corpus callosum. eEPSCs were generated by a DS3 Isolated Current Stimulator (Digitimer, Ft. Lauderdale, FL) every 20 s and stimulus intensity was adjusted to produce stable eEPSCs of 200–400 pA in amplitude prior to the initiation of experimental recording. Data were acquired using Clampex 10.3 (Molecular Devices, Sunnyvale, CA). Series resistance was monitored and only cells with a stable series resistance (less than 25 MΩ and that did not change more than 15% during recording) were included for data analysis.

Recordings were made 2–7 h after killing. The experimenter was not blinded to treatments administered to the mice.

**Experimenter-administered EtOH exposure**. Mice (PND 28–42) received i.p. injections of 0.9% NaCl saline or 2 g/kg EtOH. At 24 h after injection, mice were killed and brain slices were obtained for electrophysiological recordings (as described above). Mice were randomly selected for injection type.

**Drinking-in-the-dark model**. The DID model is a simple drinking paradigm that requires no previous training or fluid restriction[35]. Briefly, mice were offered limited access (2–4 h) to EtOH (20% v/v in water) or sucrose (2% w/v in water) via modified double ball bearing sipper tubes. Mice were randomly assigned to each fluid group. Common utilization of this model is to offer 3 days of 2 h access to these solutions, with a 4 h session on day 4 to provide an opportunity for greater consumption[35]. Mice had ad libitum access to their standard water bottles at all other times. Given that natural rewards (e.g., sugar, sex, etc.) can involve classical reward circuitry which has been studied in drug and alcohol research[70], sucrose was used in the current study as an alternative reinforcer comparison. Historically, 20% EtOH consumption via DID produces significant behavioral intoxication and blood EtOH concentrations in excess of 100 mg/dl[35]. Given that EtOH consumption occurs within this discrete period of time (2–4 h), DID is often referenced as a 'binge-like' drinking model.

In a subset of mice not included in electrophysiological experiments, retro-orbital blood samples were taken after the final DID drinking period and BECs were determined via gas chromatography (Shimadzu GC-2010 plus).

**Viral injections**. Male ChATCre+ and MOR-flox/ChATCre+ mice were anesthetized with isoflurane and stereotaxically injected with the adeno-associated viral (AAV) vector, AAV9.EF1a.DIO.ChR2(H134R)-YFP (Penn Vector Core), to drive the photosentive cation channel, ChR2, expression solely in CINs. Bilateral injections were made into the DS at coordinates anteroposterior (A/P): +0.7, mediolateral (M/L): ±1.5, dorsoventral (D/V): −3.1 (100 nl/injection, 25 nl/min infusion rate). Mice were allowed to recover for at least 2 weeks before brain slices were made for electrophysiological recordings. Male C57BL/6J mice were anesthetized with isoflurane and stereotaxically injected with the AAV vector, AAV9.hSyn.ChR2 (H134R)-eYFP (Penn Vector Core), to drive ChR2 expression in mPFC, OFC, and anterior insular neurons. Bilateral injections were made into mPFC at coordinates A/P: +1.9, M/L: ±0.3, D/V: −2.3 (100 nl/injection, 25 nl/min infusion rate); OFC: A/P: +2.7, M/L: ±1.75, D/V: −2.25 (100 nl/injection, 25 nl/min infusion rate); and anterior insular cortex: A/P: +2.4, M/L: ±2.2, D/V: −2.25 (50 nl/injection, 12.5 nl/min infusion rate).

To produce anterior insular cortical projection neuron MOR knockout mice, MOR-flox mice were anesthetized with isoflurane and stereotaxically injected with AAV9.hSyn.Cre.YFP (Penn Vector Core). Bilateral injections were made into anterior insular cortex at coordinates A/P: +2.4, M/L: ±2.2, D/V: −2.25 (50 nl/injection, 12.5 nl/min infusion rate). MOR-flox mice were allowed to recover for at least 8 weeks to allow for adequate ablation of MOR expression before brain slices were made for electrophysiological recordings.

**Optogenetic recordings**. Ai32-Emx1Cre and Ai32-Vglut2Cre transgenic mice congenitally express ChR2 on cortical and thalamic projection neurons, respectively[28,29]. AAV-ChR2-injection in ChATCre and MOR-flox/ChATCre mice allows for targeted recombination manipulations only within CINs[43,44]. They were used in the present study to express ChR2 into dorsal striatal CINs. AAV-ChR2 injection in C57BL/6J mice was performed to target ChR2 expression on inputs from mPFC, OFC, and anterior insular cortex to DLS. oEPSCs in MSNs were evoked in brain slices using 470 nm blue light (5 ms exposure time) delivered via field illumination through the microscope objective. Light intensity was adjusted to produce stable oEPSCs of 200–400 pA amplitude prior to experimental recording in Ai32 mice. In ChATCre mice, oEPSC amplitudes were generally much smaller, even with maximal light stimulation. In these recordings oEPSC amplitudes ranged from ~20 pA to 300 pA. oEPSCs were evoked once per min (AAV injected mice) or every 30 s (Ai32-Emx1Cre, Ai32-Vglut2Cre, and C57BL/6J). Prior to recording, brain slices were imaged via an Olympus MVX10 microscope (Olympus Corporation of America) to verify YFP-tagged ChR2 expression in injected ChATCre +, MOR-flox/ChATCre, and C57BL/6J mice or properly localized ChR2 expression (indicated by GFP fluorescence) in Ai32-Emx1Cre and Ai32-Vglut2Cre mice.

**Quantitative polymerase chain reaction**. Prefrontal cortex (including medial prefrontal, orbitofrontal, and anterior insular cortices) and thalamus tissue were taken from MOR-flox/Emx1cre (+) and MOR-flox/Emx1cre (−) mice. RNA was isolated from brain tissue using the RNeasy Plus Universal Mini Kit (Qiagen #73404) according to the manufacturer's protocol. Total RNA (50 ng/μl) was converted to complementary DNA (cDNA) using the High Capacity cDNA Reverse Transcription kit (Applied Biosystems Inc. (ABI), Foster City, CA: 4368814) and amplified using a TaqMan QuantStudio 6 Flex Real-Time PCR System. The TaqMan probe used in the current study was Oprm1 (Assay ID: Mm01188089_m1, Catalog #4331182, ThermoFisher). Quantitative PCR was performed using TaqMan Gene Expression Master Mix reagents (Applied

Biosystems). The relative amount of each transcript was determined via normalization across all samples to the endogenous control GAPDH (TaqMan Rodent GAPDH Control Reagents, Catalog #4308313) to account for variability in the initial concentration and quality of the total RNA and in the conversion efficiency of the reverse transcription reaction as recommended by ABI. In addition, before initiation of the analysis, cDNA was diluted 1:50 and amplified using the respective TaqMan probes to ensure that the amount of cDNA used was in the linear range. RNA samples from each individual animal were run in triplicate.

To quantify the relative expression levels of the different genes for each mouse genotype, we calculated the difference ($\Delta Ct$) between the cycle threshold of Oprm1 and the housekeeping gene glyceraldehyde 3-phosphate dehydrogenase (GAPDH). From these data, the $\Delta \Delta Ct$ ([$\Delta CtOprm1(Cre+)-\Delta CtOprm1(Cre-)$]) was computed and converted to a relative quantitative (RQ) value using the formula $2^{-\Delta \Delta Ct}$. The results were tabulated as mean ± SEM and compared between genotypes via unpaired Student's $t$ tests.

**Reagents**. The MOR agonist DAMGO (H-2535, Bachem), MOR antagonist CTAP (H-3698, Bachem), delta opioid receptor agonist [D-Pen$^2$,D-Pen$^5$]-enkephalin (DPDPE; H-2905m, Bachem), $\alpha$7 nicotinic acetylcholine receptor antagonist methyllycaconitine citrate (MLA; 1029, Tocris), AMPA antagonist NBQX (0373, Tocris), tetrodotoxin citrate (ARCD-0640, ARC Inc.), EtOH (E7148, Sigma-Aldrich), and GABA$_A$ receptor antagonist picrotoxin (P1675, Sigma-Aldrich) were used.

**Sample size**. The target number of samples in each group for our electrophysiological experiments was determined based on findings reported in our previously published studies[2]. Using these effect sizes and an $\alpha$-level set at 0.05 and at 80% power, we determined that 5–7 electrophysiological recordings from at least 2 mice was an appropriate sample size. We did not use a specific methodology for determining sample size for the qPCR analysis, but due to technical constraints determined a priori that three biological replicates would be performed.

**Replication**. All sample sizes indicated in figures for behavioral and electrophysiological experiments represent biological replicates. $N$ equals the number of slices recorded. The qPCR data are biological replicates.

**Data analyses**. Unless otherwise indicated, data were presented as the mean ± SEM. The analyses of normally distributed data were performed using two-tailed unpaired, two-tailed paired Student's $t$ tests following an F-test to confirm similar variances. Non-normally distributed data were analyzed using two-tailed Wilcoxon matched-pairs signed rank tests for paired data. Data that were analyzed using this test are indicated in the figure legends. Two-way analysis of variance (ANOVA) or linear regression were performed in Fig. 3 and supplementary Figure 2, respectively, and these data were not tested for normality. Statistical analyses were performed with Prism 7 (GraphPad, La Jolla, CA). The level of significance was set at $P < 0.05$ for all analyses. Representative traces are the average baseline EPSC (1–10 min) and average post-treatment EPSC of final 10 min of recording, unless otherwise indicated. Exclusion of individual data points was determined using an outlier calculator included in the Prism 7 software package.

**Data availability**. All data are available from the authors upon reasonable request.

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

## Acknowledgements

This work was supported by the NIH grant R00 AA023507 (B.K.A.) and institutional funds from Indiana University Health and Stark Neurosciences Research Institute. The authors would like to thank Dr. Eric Engleman for assistance with measurements of blood EtOH concentrations, and Dr. Jennifer Whistler for the generous donation of the MOR-flox mice. All other reagents and materials used for this work are commercially available.

## Author contributions

B.M., B.M.F., F.Y., and B.K.A. designed experiments, discussed the results, and contributed to all stages of manuscript preparation and editing. B.M., B.M.F., and F.Y. performed experiments. All authors revised and approved the final version of the manuscript.

## Additional information

**Competing interests:** The authors declare no competing interests.

