## [Peer Review File(PDF 304 kb) · Nature Communications]

Reviewers' comments:

Reviewer #1 (Remarks to the Author):

In this manuscript, Muñoz et al. report the results of an elegant series of studies examining the subregional and synapse-specific effects of alcohol exposure on mu opioid receptor-mediated long-term depression in dorsal striatum. Overall, these studies are well-conducted and the results represent a valuable contribution to the field. I have a few minor comments that I hope will help strengthen the impact of the manuscript.

1. The abstract could be revised to better integrate the results of the experiments and relate them to one another. This is made challenging by the fact that the experiments address 2 separate, but related questions: 1) Which glutamatergic synapses express mOP-LTD? and 2) What is the impact of alcohol exposure on mOP-LTD at these synapses? Ideally, the abstract would more clearly state how the answers to the second question inform the first. For example, the authors identify a novel form of mOP-LTD affecting CIN oEPSCs, which could drive mOP-LTD at corticostriatal synapses, except that the latter and not the former is occluded by alcohol exposure. As written, the abstract reads more as a list of results.
2. The comparison to ethanol exposure effects on mOP-LTD in DLS versus DMS should really be assessed using a 2-way ANOVA – if the data are assessed in this manner do the authors find an interaction of the effects of ethanol exposure and subregion?
3. On a related note, do the authors have any hypotheses as to why this ablation occurs specifically in DLS and not DMS? Might the weaker mOP-LTD in DMS at baseline make it more difficult to detect the effects of ethanol exposure?
4. The data from the voluntary binge drinking study are a nice addition to the paper. It is interesting that the impact of alcohol consumption is observed just 1 hour after consumption. Do the authors expect that they would see prevention of LTD at 1 hour after the single ethanol exposure from the first experiment?
5. Was there any relationship between the amount consumed in the DID procedure and subsequent mOP-LTD?
6. What statistics were used to determine the presence or lack of LTD in the experiments depicted in Figures 1 and 2?
7. While they mention the role of this circuitry in a broad array of psychological functions, the authors discuss the role of mOP effects in DLS specifically in terms of habit formation. Yet DLS mOP signaling has been shown to impact other aspects of reward-processing as well, including incentive salience (see DiFeliceantonio and Berridge, 2016).

Reviewer #2 (Remarks to the Author):

Overall the authors provide a very interesting analysis of ethanol effects on opioid receptor effects on multiple inputs to MSNs within the dorsal striatum. Within the manuscript there are several intriguing mechanisms for ethanol and opioid receptor generated LTD. However, the manuscript is disjointed and the only clear overlap is effects on opioid receptor mediated ethanol inhibition of LTD. Furthermore, some experiments only display weak evidence for what the authors are concluding, such as the CIN experiments.

Specific comments are found below:

1. Please provide representative images of ChR2 injection sites, expression, and representative terminal expression for all experiments, not just insular injections.
2. The effect of sucrose alone on synaptic plasticity is unclear. For instance, sucrose may produce a potentiation and dOR "LTD" could be depotentiation, not necessarily utilizing the same mechanisms as LTD. Therefore, the authors should use a water control as the main comparison and refer to the naturalistic control.
3. It is unclear why the paper moves from a self-administration model (DID) to single in vivo injections other than the fact that it appears the effect is independent of the type of administration. Still, it is unclear whether self-administration would produce the same effect as an acute injection on unstudied pathways such as the insular cortex.
4. The CIN experiments are intriguing. While it is obvious stimulation of CINs produces a rapid EPSC, it is unclear if this is a glutamatergic EPSC or if it is a direct effect of glutamate release from CINs. To begin to clarify this result, the authors should wash on glutamatergic antagonists to attempt to block the current. Secondly, glutamatergic terminals express nicotinic receptors which are capable of activating the terminals and releasing glutamate. Nicotinic antagonists should be used to show it is not necessary for the effect.
5. It would be interesting to know if mOR LTD is produced by ethanol exposure by occluding the mOR LTD. Therefore, a behavioral experiment which produces mOR LTD to block ethanol administration would aid in convincing the reader ethanol administration requires this effect.

Reviewer #3 (Remarks to the Author):

This manuscript presents experiments aimed at identifying the specific neural pathways within the striatum that exhibit plasticity induced by opioid receptor activation, and how such actions are altered by EtOH exposure. The authors use various viral/genetic approaches to enable selective activation or blockade of specific afferent or internal circuits within the striatum, and then use patch-clamp recording from neurons within subregions of the striatum (dorsolateral and dorsomedial) to determine how excitatory synaptic transmission in the respective pathways is altered by acute application of opioid subtype

selective ligands, and how such alterations are modulated by prior in vivo exposure to EtOH.

The authors report the following:

- 1) In coronal slices of striatum, bath application of a mu-opioid receptor (mOPR) agonist (DAMGO) induces long term synaptic depression (LTD) of local electrically-evoked excitatory postsynaptic currents (eEPSCs) in the dorsolateral and dorsomedial striatum (DLS and DMS respectively).
- 2) DAMGO induces LTD of light-evoked EPSCs from Channel Rhodopsin (ChR2) expressing cortical afferents to both DLS and DMS, but only induces short-term depression (STD) of light-evoked EPSCs from ChR2 expressing thalamic inputs to DMS and DLS.
- 3) A single in vivo exposure to systemic EtOH (2g/kg) prevents DAMGO-induced LTD of eEPSCs in the DLS but not DMS, when studied in striatal slices 24 hours after in vivo EtOH exposure.
- 4) Voluntary EtOH binge-like consumption (for 2-4 hours in a drinking in the dark paradigm) over a series of 4 days also prevented DAMGO-induced LTD of eEPSCs in the DLS that started as early as 1 hour after cessation of EtOH consumption and lasted for 3 days, with recovery to normal DAMGO-induced LTD 4 days after voluntary consumption. Importantly, voluntary consumption of sucrose, using the same protocol did not affect DAMGO-induced LTD of DLS eEPSCs.
- 5) Light-evoked DLS eEPSCs, evoked via optical stimulation of ChR2 expressing cholinergic interneurons also exhibited DAMGO-induced LTD, and the outcome that was not affected by prior in vivo EtOH exposure.
- 6) Presumed selective knockdown of mOP receptors in cortical afferents prevented DAMGO-induced LTD of eEPSCs.
- 7) Using local injections of virus driven ChR2 into cortical subregions, followed by testing of DAMGO on light-evoked EPSCs mediated by afferents from the various cortical subregions demonstrated that DAMGO only affected light-evoked responses evoked by stimulating afferents from the agranular insular cortex (where it induced LTD), but not the prefrontal, orbitofrontal or motor (previous publication) cortex.
- 8) DAMGO-induced LTD of light-induced responses from the agranular insular cortex were blocked by a mOP receptor antagonist and by prior in vivo exposure to EtOH.
- 9) Selective knockout of mOP receptors in the agranular insular cortex afferents prevented DAMGO-induced LTD of non-specific electrically evoked EPSCs, but not tetanization induced LTD of the same pathway. The manipulation also did not prevent delta-opioid induced LTD of eEPSCs.

Based on these observations the focus of the authors' conclusions are that EtOH-induced loss of opioid signaling within the striatum is mediated primarily by a very specific cortical afferent system, in particular to the DLS. Specifically, the agranular insular cortical excitatory afferent input specifically to the DLS is the primary site where acute actions of mOPR activation (LTD) is lost after contingent or non-contingent EtOH exposure in vivo.

The topic of this manuscript, neural plasticity mediating addictive responses to drugs of abuse, is clearly a very important topic, with EtOH and opioid abuse being primary preventable causes of much societal suffering and economic burden. The results and

conclusions significantly improve our understanding of the specific neural pathways that mediate key addiction related processes. More broadly, the dissection of key pathways between the cortex, thalamus and striatum, and their selective modulation by the endogenous opioid system should be of interest to the broad neuroscience community.

The design, execution and presentation of experiments is logical and clear. Although the writing is a bit choppy in places (e.g. the abstract reads a bit like a list of results more than a summary), overall the writing is clear and flows logically. Generally, I find the authors' interpretation of the experiments and their resultant conclusions to be sound. I also appreciated the authors' more expansive than usual discussion, which put their rather complex mechanistic study into a broader context.

Overall, this is a very elegant study on an important topic. However, I do think there that there are two additional experiments the authors should conduct that will improve our understanding of the mechanisms that mediate their observed outcomes (see specific comments for details). There are also a couple of presentation issues and other minor questions that should be addressed.

Specific issues/comments:

Addressing both of the following comments would significantly improve our mechanistic understanding of this study, and both should be fairly straight forward for the authors to implement.

1) Are in vivo manipulations (EtOH exposure) that prevent opioid-induced LTD in slices via occlusion, i.e. that the in vivo EtOH exposure has already maximally suppressed relevant synapses? The authors should conduct stimulus-intensity response curves to see if EtOH treated animals have reduced EPSC amplitudes relative to controls. Importantly, although the authors suggest that a past study (Wilcox et al. 2014) showed no difference in basal glutamatergic transmission after experiencing a DID protocol, it does not seem like a fair comparison. First, their model was repeated rounds of DID over weeks, so could very well have induced different effects. Second, they only analyzed spontaneous EPSCs, which come from all synapses, and so altered basal transmission strength at the very discreet subset afferent synapses would likely be swamped by no changes at the other synapses.

2) Related to comment 1 above, it would be useful to determine if bath application of or soaking of slices in a concentration of EtOH similar to what occurs in vivo (by my calculations ~40mM and ~20mM for the in vivo injection and voluntary consumption studies respectively) induce LTD in the same regional and pathway specific manner, and does it then occlude DAMGO-induced LTD in the relevant selective manner? Or, even if it doesn't induce LTD on its own, does an EtOH exposure in vitro prevent subsequent DAMGO-induced plasticity.

Minor comments:

1) Given that cells are clamped at -60mV, which presumably means the e/oEPSCs are primarily mediated by AMPA receptors (due to voltage-dependent block of NMDARs), the e/oEPSCs generally seem a bit slow, typically 50ms duration and long slow tail. Thus, I am

concerned that the synaptic currents are not very well clamped. Accordingly, I am wondering if changes of 15% in R_i (the reported threshold for excluding an experiment) could underly LTD or STD? The idea is argued against by the fact that DAMGO induced plasticity is dependent on the afferents stimulated despite being the same post synaptic cell type. However, it would be helpful to plot R_i versus time, along with EPSC amplitude versus time, for at least some of the key experiments.

2) While it is possible I missed a specific passage or reference, it wasn't entirely clear to me what the evidence was that demonstrated that the presumed region of origin specific ChR2 afferents are actually specific.

3) The authors indicate concerns about stress effects for the injection experiments, and they have implemented some reasonable controls to reduce this possibility. However, in addition to injections and other related mechanical aspects that could cause stress, have the authors considered that EtOH consumption or withdrawal induce stress responses/hormones? Obviously, in this case the results would still be relevant, but it would be worth considering whether the induced actions are pharmacological/neurological versus a stress response, even if it is due to drug use or withdrawal.

4) I was unable to ascertain what the gray bars on day 4 of the consumption graphs in Fig 4b&c stand for? Given the apparent doubling of consumption, I am presuming the animals had access for 2 hours on day 1-3, and 4 hours on day 4, but unless I missed it, this is never explicitly described in text, methods or fig legends?

5) The section heading on page 10, line 164 is confusing. It implies that the effects of EtOH on LTD are synapse specific, but LTD is only observed at cortical synapses anyway. So, although impacts of in vivo EtOH are clearly synapse specific, one involves LTD and the other involves STD. The current wording makes it sound like LTD occurs at both synapses, but only one is prevented by in vivo EtOH. Some rewording of the section heading would be more accurate and thus helpful. Perhaps, "EtOH block of mOP-induced plasticity is specific to cortical inputs"?

6) Although the RQ-PCR data in Fig. 7a confirm that the mOP receptor is effectively reduced, I did not see any data demonstrating that the knockdown is specific to cortical afferents. This should be directly demonstrated with either immunocytochemistry, or at least by demonstrating with PCR that mOP receptor levels are not affected in the thalamus.

Authors' responses to reviewers

We greatly appreciate the time and effort the reviewers put into the evaluation of our work. Their comments and recommendations were very helpful and have made this study more rigorous and afforded us the opportunity to make a few novel discoveries that we feel will give the work an even larger impact. Our responses to the individual reviewer's comments are found below.

Reviewers' comments:

Reviewer #1 (Remarks to the Author):

1. The abstract could be revised to better integrate the results of the experiments and relate them to one another. This is made challenging by the fact that the experiments address 2 separate, but related questions: 1) Which glutamatergic synapses express mOP-LTD? and 2) What is the impact of alcohol exposure on mOP-LTD at these synapses? Ideally, the abstract would more clearly state how the answers to the second question inform the first. For example, the authors identify a novel form of mOP-LTD affecting CIN oEPSCs, which could drive mOP-LTD at corticostriatal synapses, except that the latter and not the former is occluded by alcohol exposure. As written, the abstract reads more as a list of results.

We agree that the abstract could have read more smoothly. We have now described our results more succinctly and in the context of the larger research question. We think the better contextualization and the focused description of our results improves its integration. It also matches the abstract style of the journal more closely.

2. The comparison to ethanol exposure effects on mOP-LTD in DLS versus DMS should really be assessed using a 2-way ANOVA – if the data are assessed in this manner do the authors find an interaction of the effects of ethanol exposure and subregion?

Thank you for this suggestion as it is indeed a more appropriate statistical approach for our specific question. A 2-way ANOVA indeed shows an interaction between EtOH exposure and subregion. We have added the statistics in the legend of Figure 3, page 34, line 643-645.

3. On a related note, do the authors have any hypotheses as to why this ablation occurs specifically in DLS and not DMS? Might the weaker mOP-LTD in DMS at baseline make it more difficult to detect the effects of ethanol exposure?

Specific EtOH effects could be due to the different innervations each dorsal striatal subregion receives (Hunnicuttt et al., 2017 & insular inputs reach mainly DLS, Fig. 8a), resulting in different synaptic environments. As such, this may explain differences in their sensitivity to EtOH. The lack of a statistical difference between subregions in mOP-LTD magnitude (Figure 3) also suggests that no pre-existing differences in the relative strengths of mOP-LTD in DMS versus DLS could have precluded the potential to observe an EtOH effect in the DMS. This has been added to the discussion section page 16 line 340-347. Future work will explore the molecular underpinnings between ethanol-sensitive and insensitive synapses.

4. The data from the voluntary binge drinking study are a nice addition to the paper. It is interesting

that the impact of alcohol consumption is observed just 1 hour after consumption. Do the authors expect that they would see prevention of LTD at 1 hour after the single ethanol exposure from the first experiment?

We expect the same result for an ip EtOH injection since our previous data show that an oxycodone ip injection blocks the induction of mOP-LTD 1 h after (Atwood et al., 2014) and persists for 3 days. This is very similar to the binge drinking study results. We have mentioned this in the discussion, page 14-15, lines 307-315.

5. Was there any relationship between the amount consumed in the DID procedure and subsequent mOP-LTD?

We attempted to address this and found no significant correlation between EtOH intake and mOP-LTD magnitude at 24 h, where the effect was maximal. This is likely due the fact that cells from this time point were only from 2 animals and each had consumed similar amounts of EtOH (6.3 & 6.7 g/kg). This is, however, an interesting and important question that merits future study that will require a significantly larger cohort of mice which offers much more variability in EtOH intake.

6. What statistics were used to determine the presence or lack of LTD in the experiments depicted in Figures 1 and 2?

We have included Supplementary Figure 1 which compares baseline and post-DAMGO EPSC amplitudes from Figure 1 and 2. These data demonstrate a significant decrease in EPSC amplitude following DAMGO application in both DLS and DMS from C57BL/6J and Emx1-Ai32 mice. Vglut2-Ai32 data shows the return of EPSC amplitude to baseline levels after DAMGO application. Results section, page 5 line 98 and 112, page 6 line 113, 118 and 119.

7. While they mention the role of this circuitry in a broad array of psychological functions, the authors discuss the role of mOP effects in DLS specifically in terms of habit formation. Yet DLS mOP signaling has been shown to impact other aspects of reward-processing as well, including incentive salience (see DiFeliceantonio and Berridge, 2016).

Thank you for pointing this out and it is important to include in the discussion of our results. We have altered the discussion section to be more inclusive of this additional information (page 17-18, lines 355-359 and 375-379)

Reviewer #2 (Remarks to the Author):

1. Please provide representative images of Chr2 injection sites, expression, and representative terminal expression for all experiments, not just insular injections.

These images have been added in Supplementary Figure 9.

2. The effect of sucrose alone on synaptic plasticity is unclear. For instance, sucrose may produce a potentiation and dOR “LTD” could be depotentiation, not necessarily utilizing the same mechanisms as LTD. Therefore, the authors should use a water control as the main comparison and refer to the naturalistic control.

We used sucrose as a natural reward control because in all of the other experiments, mice had access to water *ad libitum*. Therefore, all naïve experiments (Figures 1,2,7,8,9) were already water controls. Furthermore, visual examination of the relative magnitude of mOP-LTD in naïve (Figure 1) and sucrose drinking mice (Figure 4) suggests that DLS plasticity is virtually identical. We added an explanation to the Methods section clarifying that mice had *ad libitum* access to water at all other times during the DID experiments, page 23 line 475 - 480. We feel that the sucrose experiments add an additional element to the story, that our findings our ethanol-specific as a natural reward did not produce the same effect.

3. It is unclear why the paper moves from a self-administration model (DID) to single in vivo injections other than the fact that it appears the effect is independent of the type of administration. Still, it is unclear whether self-administration would produce the same effect as an acute injection on unstudied pathways such as the insular cortex.

Figs. 8 and 9 demonstrate that the ethanol-sensitive mOP-LTD we measured in all of our other figures was occurring exclusively at insular cortex inputs to DLS. Therefore, we can infer that in the DID experiment, the only synapses affected by EtOH were the insular synapses.

4. The CIN experiments are intriguing. While it is obvious stimulation of CINs produces a rapid EPSC, it is unclear if this is a glutamatergic EPSC or if it is a direct effect of glutamate release from CINs. To begin to clarify this result, the authors should wash on glutamatergic antagonists to attempt to block the current. Secondly, glutamatergic terminals express nicotinic receptors which are capable of activating the terminals and releasing glutamate. Nicotinic antagonists should be used to show it is not necessary for the effect.

Thank you for these suggestions; they are important considerations. We have added these data as Supplementary Figure 7 and Supplementary Figure 8. The Results section also includes the findings of these additional experiments, page 9 and 10 line 199-213. We found that an alpha7 nAChR antagonist (glutamate terminal-expressed nAChR) did not prevent our LTD, NBQX blocked the majority of our EPSC, and MORflox-ChATcre mice that lack MOPr in CINs did not express CIN-mOP-LTD. Altogether it appears that CIN-mOP-LTD results from activation of MOPr in CINs, reducing glutamate release directly from CINs and not indirectly from glutamate inputs to DLS. In addition, we added a brief statement in the Discussion section page 13, lines 273-276.

5. It would be interesting to know if mOR LTD is produced by ethanol exposure by occluding the mOR LTD. Therefore, a behavioral experiment which produces mOR LTD to block ethanol administration would aid in convincing the reader ethanol administration requires this effect.

Thank you for these suggestions. This is a similar point as was proposed by Reviewer #3 (points 1 and 2). We have conducted additional experiments to determine if ethanol was preventing our LTD by occlusion or by blocking its induction and the findings are presented in Supplementary Figures 3 through 6. We found that EtOH bath application neither induced nor blocked mOP-LTD at both low and high concentrations (Supplementary Fig. 3, Results section page 9, lines 182-184). In addition, in vivo ethanol exposure did not produce a change in stimulus intensity-response (Supplementary Fig. 4, Results section page 9, lines 185-186), sEPSCs (Supplementary Fig. 5, Results section page 9, lines 186-187) and mEPSCs (Supplementary Fig. 6, Results section page 9, lines 187-189). Altogether these data do not demonstrate any differences in basal glutamate transmission, suggesting LTD has not been induced. This is consistent with others that found that even drinking-in-the-dark that was much more extensive than ours did not alter basal glutamate transmission (Wilcox et al., 2014). Therefore, EtOH's effects are not through occlusion, but must engage a signaling pathway in vivo that prevents LTD induction. Future work will explore this line of reasoning.

Reviewer #3 (Remarks to the Author):

1) Are in vivo manipulations (EtOH exposure) that prevent opioid-induced LTD in slices via occlusion, i.e. that the in vivo EtOH exposure has already maximally suppressed relevant synapses? The authors should conduct stimulus-intensity response curves to see if EtOH treated animals have reduced EPSC amplitudes relative to controls. Importantly, although the authors suggest that a past study (Wilcox et al. 2014) showed no difference in basal glutamatergic transmission after experiencing a DID protocol, it does not seem like a fair comparison. First, their model was repeated rounds of DID over weeks, so could very well have induced different effects. Second, they only analyzed spontaneous EPSCs, which come from all synapses, and so altered basal transmission strength at the very discreet subset afferent synapses would likely be swamped by no changes at the other synapses.

We will consider this point in conjunction with point #2 (see below)

2) Related to comment 1 above, it would be useful to determine if bath application of or soaking of slices in a concentration of EtOH similar to what occurs in vivo (by my calculations ~40mM and ~20mM for the in vivo injection and voluntary consumption studies respectively) induce LTD in the same regional and pathway specific manner, and does it then occlude DAMGO-induced LTD in the relevant selective manner? Or, even if it doesn't induce LTD on its own, does an EtOH exposure in vitro prevent subsequent DAMGO-induced plasticity.

Thank you, these are truly important considerations and similar to a point raised by Reviewer #2 (point 5). We have conducted a number of experiments to address these questions. Stimulus-intensity response analyses did not show a difference between saline and ethanol-injected animals (Supplementary Fig. 4, Results section page 9, lines 185-186). Our previous work demonstrated that mOP-LTD does produce a measureable decrease in sEPSC frequency, but not amplitude (Atwood et al., 2014). To support the stimulus-response data we also measured sEPSCs (Supplementary Fig. 5, Results section page 9, lines 186-187), and mEPSCs (Supplementary Fig. 6, Results section page 9, lines 187-189) 24 h-post saline or EtOH exposure and also found no difference, consistent with the much more extensive DID experiments in the Wilcox et al., 2014 study. We also performed additional experiments (Supplementary Fig. 3, Results section page 9, lines 182-184) wherein 20 and 40 mM EtOH were washed onto slices and neither induced nor blocked mOP-LTD. Collectively, these additional experiments provide solid evidence that EtOH exposure does not induce or occlude mOP-LTD.

Minor comments:

1) Given that cells are clamped at -60mV, which presumably means the e/oEPSCs are primarily mediated by AMPA receptors (due to voltage-dependent block of NMDARs), the e/oEPSCs generally seem a bit slow, typically 50ms duration and long slow tail. Thus, I am concerned that the synaptic currents are not very well clamped. Accordingly, I am wondering if changes of 15% in R_i (the reported threshold for excluding an experiment) could underly LTD or STD? The idea is argued against by the fact that DAMGO induced plasticity is dependent on the afferents stimulated despite being the same post synaptic cell type. However, it would be helpful to plot R_i versus time, along with EPSC amplitude versus time, for at least some of the key experiments.

Thank you. We have plotted R_i for our initial experiments demonstrating mOP-LTD in DLS and DMS and included these graphs as panels in Fig. 1 and a brief description in the result section (page 5, lines 101-102). We extensively characterized mOP-LTD in our previous publication (Atwood et al., 2014) and the data in this study are entirely consistent with our previous work in time course and magnitude.

2) While it is possible I missed a specific passage or reference, it wasn't entirely clear to me what the evidence was that demonstrated that the presumed region of origin specific ChR2 afferents are actually specific.

The expression specificity of ChR2 in transgenic mice (Ai32/Emx1 and Ai32/Vglut2) has been supported with multiple references on page 5 line 108-110. Fig 7 also demonstrates that knockdown of MOPr from cortex (MOPr flox/Emx1Cre+) was confirmed and recordings from these mice demonstrated a functional loss as well due to a lack of mOP-LTD.

3) The authors indicate concerns about stress effects for the injection experiments, and they have implemented some reasonable controls to reduce this possibility. However, in addition to injections and other related mechanical aspects that could cause stress, have the authors considered that EtOH consumption or withdrawal induce stress responses/hormones? Obviously, in this case the results would still be relevant, but it would be worth considering whether the induced actions are pharmacological/neurological versus a stress response, even if it is due to drug use or withdrawal.

Synaptic plasticity is indeed sensitive to stress, environment, and treatment. In our case, we performed some experiments with saline-injected controls. Comparatively, mOP-LTD data from these animals are highly similar to experiments with naïve mice. In addition, the DMS data in Fig. 3 indicate that EtOH injected mice show no effect of EtOH exposure on mOP-LTD. As this was 24 h later, any sort of withdrawal state that may interfere with plasticity would be present, however, no effect was found. Furthermore, in Fig. 7b the mice were not injected and mOP-LTD was blocked by the knockdown of MOPr from the cortex, not by injection or stress. Finally, in Fig. 9 we show a disruption of mOP-LTD when we knockdown MOPr from insular cortex, however other synaptic plasticity was not affected as eCB-LTD (also affected by EtOH exposure and stress) and dOP-LTD were intact (Supplementary Fig.10). We have included a brief section in the discussion addressing this possibility (page 18, lines 380-394).

4) I was unable to ascertain what the gray bars on day 4 of the consumption graphs in Fig 4b&c stand for? Given the apparent doubling of consumption, I am presuming the animals had access for 2 hours on day 1-3, and 4 hours on day 4, but unless I missed it, this is never explicitly described in text, methods or fig legends?

Yes, you are correct that the gray bars indicate a 4 h session on day 4. We apologize that this was not clear. We have added this information to the methods (page 23, lines 475-476) and figure legend for Fig. 4 (page 36, line 662-663) and also for Supplementary Fig. 2.

5) The section heading on page 10, line 164 is confusing. It implies that the effects of EtOH on LTD are synapse specific, but LTD is only observed at cortical synapses anyway. So, although impacts of in vivo EtOH are clearly synapse specific, one involves LTD and the other involves STD. The current wording makes it sound like LTD occurs at both synapses, but only one is prevented by in vivo EtOH. Some rewording of the section heading would be more accurate and thus helpful. Perhaps, "EtOH block of mOP-induced plasticity is specific to cortical inputs"?

This has been clarified (page 8, line 172-173).

6) Although the RQ-PCR data in Fig. 7a confirm that the mOP receptor is effectively reduced, I did not see any data demonstrating that the knockdown is specific to cortical afferents. This should be

directly demonstrated with either immunocytochemistry, or at least by demonstrating with PCR that mOP receptor levels are not affected in the thalamus.

This is indeed an important consideration. We repeated our qPCR analysis in additional MOPr-flox/Emx1Cre mice and took samples from thalamus in addition to cortex for our analyses. We confirmed that MOPr expression is indeed reduced in cortex in the Cre (+) mice, but remains intact in the thalamus. We included these data in Figure 7a,b (page 41, line 694-701). These data are consistent with others that have used the Emx1cre mice for studies of corticostriatal signaling and we referenced these studies in this revised manuscript as well (Kupferschmidt et al. Neuron, 2017, Wu et al. Cell Reports 2015; Scharf et al., 2016; Garcia et al., 2015).

REVIEWERS' COMMENTS:

Reviewer #1 (Remarks to the Author):

The authors have appropriately addressed all my concerns. The revisions have strengthened what was already a valuable addition to the literature.

Reviewer #2 (Remarks to the Author):

I am satisfied with the way that the authors have addressed my comments, and added additional experiments. No further comments.

Reviewer #3 (Remarks to the Author):

The authors have done a nice job of addressing my concerns.

I have one minor grammar comment, and that is, in the revised text, line 185, the authors transition from the in vitro EtOH experiments straight into the in vivo experiments, without a helpful transition word. So, in line 185, when it says, "Furthermore, EtOH exposure did not affect...", coming right after the in vitro passage, it makes it seem like these are other lack of effects of bath applied EtOH. The authors should add in vivo, or some other clarifying transition to avoid this confusion.

Nice work!

Authors' response to reviewers.

REVIEWERS' COMMENTS:

Reviewer #1 (Remarks to the Author):

The authors have appropriately addressed all my concerns. The revisions have strengthened what was already a valuable addition to the literature.

No action requested.

Reviewer #2 (Remarks to the Author):

I am satisfied with the way that the authors have addressed my comments, and added additional experiments. No further comments.

No action requested.

Reviewer #3 (Remarks to the Author):

The authors have done a nice job of addressing my concerns.

I have one minor grammar comment, and that is, in the revised text, line 185, the authors transition from the in vitro EtOH experiments straight into the in vivo experiments, without a helpful transition word. So, in line 185, when it says, "Furthermore, EtOH exposure did not affect...", coming right after the in vitro passage, it makes it seem like these are other lack of effects of bath applied EtOH. The authors should add in vivo, or some other clarifying transition to avoid this confusion.

We have added the words "in vivo" to the statement indicated above. It now reads "Furthermore, in vivo EtOH exposure did not affect..."